# The Influence of Bisphenol A (BPA) on the Occurrence of Selected Active Substances in Neuregulin 1 (NRG1)-Positive Enteric Neurons in the Porcine Large Intestine

**DOI:** 10.3390/ijms221910308

**Published:** 2021-09-24

**Authors:** Krystyna Makowska, Kamila Szymańska, Jarosław Całka, Sławomir Gonkowski

**Affiliations:** 1Department of Clinical Diagnostics, Faculty of Veterinary Medicine, University of Warmia and Mazury in Olsztyn, Oczapowskiego 14, 10-957 Olsztyn, Poland; 2Department of Clinical Physiology, Faculty of Veterinary Medicine, University of Warmia and Mazury in Olsztyn, Oczapowskiego 13, 10-957 Olsztyn, Poland; kamila.szymanska@uwm.edu.pl (K.S.); jaroslaw.calka@uwm.edu.pl (J.C.); slawomir.gonkowski@uwm.edu.pl (S.G.)

**Keywords:** Neuregulin 1, enteric nervous system, large intestine, bisphenol A

## Abstract

Bisphenol A (BPA) is a substance used in the manufacture of plastics which shows multidirectional adverse effects on living organisms. Since the main path of intoxication with BPA is via the gastrointestinal (GI) tract, the stomach and intestine are especially vulnerable to the impact of this substance. One of the main factors participating in the regulation of intestinal functions is the enteric nervous system (ENS), which is characterized by high neurochemical diversity. Neuregulin 1 (NRG1) is one of the lesser-known active substances in the ENS. During the present study (performed using the double immunofluorescence method), the co-localization of NRG1 with other neuronal substances in the ENS of the caecum and the ascending and descending colon has been investigated under physiological conditions and after the administration of BPA. The obtained results indicate that NRG1-positive neurons also contain substance P, vasoactive intestinal polypeptide, a neuronal isoform of nitric oxide synthase and galanin and the degree of each co-localization depend on the type of enteric plexus and the particular fragment of the intestine. Moreover, it has been shown that BPA generally increases the degree of co-localization of NRG1 with other substances.

## 1. Introduction

Bisphenol A (BPA) is an organic synthetic substance which is used in the manufacture of plastics around the world and therefore contained in a wide range of objects of everyday use, including domestic appliances, toys, bottles, conserve tin liners, office supplies, dental materials, amongst many other things [1,2]. BPA may leach out from plastics and penetrate food, water, soil and air [3]. Moreover, it is known that BPA may penetrate living organisms through the gastrointestinal tract, respiratory system or the skin [1]. Due to its similarity to oestrogen, BPA has the ability to bind to receptors for this hormone and therefore may have adverse effects on many internal organs [4]. Based on previous studies, it is known that BPA affects several systems, including the nervous, reproductive, immunological, endocrine and excretory systems [1,2].

However, it should be pointed out that the main vector of intoxication of living organisms with BPA is through the gastrointestinal (GI) tract [5]. Therefore, the stomach and intestine are particularly vulnerable to the harmful effects of this substance. Previous studies have reported that BPA may first of all affect the mucosal layer of the GI tract, causing the intensification of apoptosis and slowdown of processes connected with the proliferation of intestinal epithelial cells [6]. As a result, BPA evokes inhibition of the secretion of mucin by intestinal epithelial cells [7] and affects the intestinal barrier, resulting in an increase in intestinal permeability [8]. Moreover, it is known that BPA clearly influences intestinal motility, leading to the relaxation of the smooth muscles [9]. One of the more interesting aspects of BPA impact on the GI tract is the influence of this substance on a specific part of the nervous system, i.e., the enteric nervous system (ENS) [10,11]. However, many aspects concerning the impact of BPA on enteric neurons remain unclear.

The ENS is located in the wall of the GI tract and is characterized by a very complicated construction and a high degree of independence from the central nervous system [12,13]. The ENS is composed of millions of neurons grouped in neuronal ganglia interconnected with a dense network of nerve fibres forming enteric plexuses [12,13]. The number and localization of enteric plexuses depend on the mammal species in question. In rodent intestines, two types of enteric plexuses are present, i.e., the myenteric plexus (MP) in the intestinal muscular layer and submucous plexus, near the lamina propria of the mucosa [14]. In turn, in large mammal species and humans, the submucous plexus is divided into two plexuses: the inner submucous plexus, located in the same place as the submucous plexus in rodents, and the outer submucous plexus, located near the side of the circular muscle layer [15].

The enteric neurons are characterized by a high degree of heterogeneity with regard to their neurochemical characterization, which enables the synthesis of active substances by neuronal cells [12,16]. Besides acetylcholine—the main classic neuromediator of the enteric neurons—several dozen other neuronal substances have been found within the ENS [16]. It should be underlined that the distribution of the enteric nervous structures and the exact functions of some of them are not fully understood. One of the lesser-known substances occurring in the ENS is neuregulin 1 (NRG1).

NRG1 is a 44-kD glycoprotein belonging to the neuregulin family, which, in turn, is a part of the group of epithelial growth factors affecting ErbB receptors, and plays a key role in the development of the organism through the regulation of apoptosis, angiogenesis and cell survivability [17,18]. In the central nervous system, NRG1 is involved in the proper differentiation and functioning of oligodendrocytes [19] and regulates the higher nervous functions [20]. In turn, in the peripheral nervous system, NRG1 affects the Schwann cells, showing neuroprotective activity, and influences the ion channels located in the neuronal membrane and the activity of other neuronal active substances [21,22]. It should be underlined that the current knowledge of NRG1 in the ENS is extremely sparse. However, it is known that NRG1 is present in the enteric nervous structures of some mammal species, including humans [23,24], and that the number of NRG1-positive nervous structures supplying internal organs may undergo changes in response to exposure to BPA [25], though the exact functions of this substance in intestinal innervation remains unknown.

One of the ways to better understand the functions of lesser-known neuronal substances is the study of their co-localization with other better-known factors in the same nervous structures, since previous investigations have shown that neuronal substances synthesized and occurring in the same neurons most often play similar and/or complementary roles [26,27,28]. Therefore, the aim of the present study was to determine the degree of co-localization of NRG1 with other selected neuronal factors whose roles are better understood in the regulation of intestinal functions. The research included substance P (SP), galanin (GAL), vasoactive intestinal polypeptide (VIP) and a neuronal isoform of nitric oxide synthase (nNOS—used as a marker of neurons synthesizing nitric oxide (NO). These substances are known as regulatory factors and are involved in various regulatory processes of intestinal activity, both under physiological conditions and during pathological states and/or intoxications [10,27,29]. SP is primarily a factor that takes part in sensory conduction and induces intestinal motility [16,30], while VIP and NO are the inhibitory factors reducing intestinal motility and secretion [31,32]. Furthermore GAL regulates the synthesis of other enteric neuropeptides, however its precise functions depends on the intestinal fragment and mammal species in question [33,34]. Moreover, it is known that all of the above-mentioned substances perform adaptive and protective functions under the impact of pathological factors [10,29]. Therefore, the identification of the neurochemical characterization of NRG1-positive enteric neurons and the degree of co-localization of NRG1 with other better-known substances will expand knowledge of the functions of NRG1 in the ENS of the large intestine in physiological conditions and under the impact of BPA. Moreover, due to neurochemical and anatomical similarities between the human and porcine ENS [35]—which is what makes the domestic pig a good animal model for studies of processes in human intestinal innervation—the obtained results may be the first step toward understanding the functions of NRG1 in the human intestine.

## 2. Results

During this study, all neuronal factors studied were found in NRG1-like immunoreactive (LI) neurons located in all types of enteric plexus within the porcine caecum and the ascending and descending colon both under physiological conditions and after BPA administration. The degree of co-localization of NRG1 with other substances depended on the type of enteric plexus and the fragment of the intestine.

### 2.1. Caecum

Under physiological conditions, the largest number of NRG1-positive cells in all types of enteric plexus were also immunoreactive to nNOS (Table 1), which was found in 31.67 ± 0.73%, 32.06 ± 0.58% and 30.46 ± 0.21% of all neurons immunoreactive to NRG1 in the MP, OSP and ISP, respectively. A clearly lower degree of co-localization was noted in the case of NRG1 and VIP. The percentage of NRG1–LI neurons, which simultaneously contained these two substances ranged from 24.93 ± 0.30% in the ISP to 26.39 ± 0.29% in the OSP (Table 1). A lower percentage of NRG1-positive cells were also positive for SP, which was present in 21.05 ± 0.38%, 23.47 ± 0.37% and 22.14 ± 0.42% of all neurons immunoreactive to NRG1 in the MP, OSP and ISP, respectively (Table 1). In turn, GAL was observed in the smallest number of NRG1–LI neurons. The percentage of NRG1+/GAL+ observed ranged from 19.94 ± 0.20% of all NRG1-positive neurons in the MP to 22.00 ± 0.31% in the ISP (Table 1).

Administration of low and high doses of BPA caused an increase in the degree of co-localization of NRG1 with all other substances studied in all types of enteric plexus, and more visible changes were observed with higher doses of BPA (Table 1, Figure 1). The clearest changes were noted in the percentage of NRG1+/GAL+ neurons located in the MP and in the percentage of NRG+/SP+ neurons in the ISP. In the former, the administration of a high dose of BPA caused an increase in the percentage of such neurons from 19.94 ± 0.20% to 35.44 ± 0.18% of all NRG1–LI cells, and in the latter from 22.14 ± 0.42% to 38.05 ± 0.23%. In both cases, the percentage of neurons increased by about 16 percentage points (pp) compared to the control animals. The administration of BPA also caused significant changes (about 15 pp compared to control animals) in the population of neurons NRG1+/VIP+ located in the ISP (the increase to 39.09 ± 0.12% of all NRG1–LI neurons) and NRG1+/SP+ in the MP (the increase to 36.59 ± 0.34%)

The changes on the level of about 10 pp in comparison to the control animals were also observed under the impact of a high dose of BPA in populations of neurons NRG1+/VIP+ in the MP and NRG1+/SP+ in the OSP, as well as in animals treated with a low dose of BPA in neurons simultaneously immunoreactive to NRG1+/SP+ located in the ISP (Table 1). In turn, less visible changes (only about 2 pp in comparison to control animals) were noted under the impact of a low dose of BPA in the population of neurons NRG1+/nNOS+ located in the OSP and ISP (Table 1). Interestingly, in the ISP of animals treated with a high dose of BPA, the largest number of NRG1-positive neurons was found in co-localization with VIP (39.09 ± 0.12% of all NRG1–LI cells) and/or SP (38.05 ± 0.23%) (in contrast to the control animals, for which NRG1+/nNOS+ were most numerous) (Table 1).

### 2.2. Ascending Colon

Under physiological conditions in the MP and ISP, the degree of co-localization of NRG1 with other substances studied was similar to that observed in the caecum (Table 2, Figure 2). The most numerous of NRG1–LI neurons were also immunoreactive to nNOS, which was found in 32.80 ± 0.84% of all NRG1-positive neurons in the MP and in 31.52 ± 0.49% in the ISP.

Less numerous were NRG1+/VIP+ and NRG1+/SP+ neurons, whereas the smallest degree of co-localization was observed for neurons immunoreactive to NRG1 and GAL, whose percentage did not exceed 20% of NRG1-positive cells in each type of enteric plexus (Table 2). The situation was slightly different in the OSP, where the largest number of neurons immunoreactive to NRG1 also showed the presence of VIP, which was noted in 35.76 ± 0.36% of all NRG1–LI neurons. In this type of plexus, NRG1+/nNOS+ neurons (unlike in other types of enteric plexus) formed the second-highest population in terms of numbers (Table 2).

The administration of BPA generally caused, likewise in the caecum, an increase in the degree of co-localization of NRG1 with other substances studied. The only exception was the number of NRG1+/nNOS+ neurons in the OSP, which did not undergo statistically significant changes under the impact of a low dose of BPA (Table 2). In turn, the most visible changes concerned NRG1+/VIP+ neurons in the ISP, whose number increased from 21.39 ± 0.23% in control animals to 36.09 ± 0.40% in pigs treated with low doses of BPA (by about 15 pp) and to 45.74 ± 0.24% in animals after administration of high dose of BPA (by about 24 pp) (Table 2). The visible impact of BPA was also evident for NRG1+/SP+ neurons located in the OSP and ISP, where a high dose of BPA resulted in the increase from 22.12 ± 0.33% to 39.64 ± 0.29% (by about 17 pp) and from 21.82 ± 0.43% to 36.03 ± 0.13% (by about 15 pp), respectively. Similar changes (an increase by about 15 pp in comparison to the control animals) were also caused by a high dose of BPA in NRG1+/GAL+ neurons located in the MP and OSP (Table 2).

### 2.3. Descending Colon

Under physiological conditions, the degree of co-localization of NRG1 with all other substances studied in the enteric neurons located in the descending colon was higher than in other parts of the large intestine (Table 3, Figure 3).

This phenomenon was particularly pronounced in the case of NRG1+/GAL+ neurons, which formed the largest population of NRG1–LI neuronal cells in the OSP and ISP (Table 3). In these kinds of enteric plexuses, GAL was found in 38.33 ± 1.06% and 38.70 ± 1.31% of all NRG1–LI neurons, respectively. In the OSP and ISP, a slightly smaller percentage of NRG1–LI neurons also showed immunoreactivity to nNOS and GAL, and the least numerous were neurons exhibiting the co-localization of NRG1 and SP (Table 3). In turn, within the MP (likewise in the caecum and the ascending colon) the largest number of NRG1-positive neurons were also immunoreactive to nNOS, which was noted in 34.17 ± 1.66% of all cells containing NRG1 (Table 3).

In the descending colon, low doses of BPA caused the least pronounced changes among all segments of the large intestine studied. Such doses did not result in statistically significant changes in the percentage of NRG1+/SP+ neurons located in all types of the enteric plexus, NRG1+/nNOS+ neurons in the MP and OSP, or in NRG1+/VIP+ cells in the ISP (Table 3). In turn, the most visible changes caused by low doses of BPA concerned the percentage of NRG1+/GAL+ neurons located in the MP (an increase from 30.14 ± 0.69% to 37.71 ± 0.34%, by about 7 pp) and OSP (the increase from 38.33 ± 1.06% to 44.26 ± 1.14%, by about 6 pp) (Table 3).

High doses of BPA in the descending colon (similar to the caecum and ascending colon) caused an increase in the degree of co-localization of NRG1 with all other substances studied in all types of enteric plexus (Table 3). The most visible changes were noted in the MP and concerned NRG1+/GAL+ neurons (an increase from 30.14 ± 0.69% to 47.20 ± 0.91%, by about 17 pp) and NRG1+/VIP+ neurons (an increase from 27.94 ± 1.82% to 42.92 ± 1.90%, by about 15 pp) (Table 3).

## 3. Discussion

During the present study, NRG1 was observed in the enteric neurons located in all types of enteric plexus within the porcine large intestine, which is in agreement with previous studies in which this substance was noted in the ENS of other mammal species, including humans [23,24]. NRG 1 has also been described in the porcine ENS [36]. The distribution of NRG1-positive neurons in all types of enteric plexus observed in the present study and previous investigations [36] strongly suggest that this substance has multiple functions in the intestine, including the regulation of intestinal motility (mainly regulated by the myenteric plexuses) and secretory activity (for which submucous plexuses are responsible) [12,13,37]. The wider distribution of NRG1 in the ENS may also be connected with the participation of this substance in the development and growth of enteric neurons as these processes are understood from previous studies [24,38], as well as in protective action on Schwann cells [39]. Previous studies have described BPA as affecting the total number of NRG1-positive neurons in the ENS [36]. In turn, the present experiment, for the first time, shows the neurochemical characterization of the enteric neurons immunoreactive to NRG1, and clearly indicates that this substance may co-localize in the same neurons with various other neuronal active substances.

In the light of the present study, the relatively large number of NRG1-positive enteric neurons also showed the presence of nNOS and/or VIP, which may suggest that NRG1 (similar to both of these substances) may perform inhibitory functions in the large intestine. Such a conclusion follows from the fact that both VIP and NO are the two most important intestinal inhibitory factors. They affect the smooth muscles, leading to hyperpolarization of the muscular fibres and therefore to inhibition of intestinal motility [32,40,41]. The participation of NRG1 in intestinal motility (indirectly confirmed in the present study by co-localization of this substance with VIP and nNOS) is very likely due to the fact that previous studies have shown that NRG1 is involved in the regulation of muscular metabolism and affects the neuro-muscular synapses [42]. Moreover, VIP and NO influence the secretory activity of the GI tract [40,41] and show vasodilatory effects on the blood vessels located in the intestinal wall and mesentery, and the co-localization of these substances with NRG1 may indicate that NRG1 also affects intestinal blood vessels.

In turn, the co-localization of NRG1 with SP, which is an important factor participating in sensory conduction, may also indicate that NRG1 displays such activity and may occur in the sensory enteric neurons. This thesis is all the more probable since previous studies have suggested the participation of neuronal NRG1 in the conduction of sensory and/or pain stimuli [43]. Moreover, it is known that SP is an important factor which regulates intestinal motility and secretion [44,45]. Contrary to VIP and NO which, as mentioned above, have clear inhibitory effects on the gastrointestinal muscles [40,41], the impact of SP on intestinal motility is not uniquely defined. The previous studies have shown that SP may cause both contraction and relaxation of the intestinal muscles, the character of the impact depending on the type of receptor stimulated and on the particular species of mammal. It is known that stimulation of neurokinin-3 receptors results in the stimulation of intestinal motility, whereas stimulation of neurokoinin-1 receptors has relaxatory effects [46]. It is also known that SP in the rat GI tract shows strong contractile effects, whereas in humans such activity is very limited [47,48].

GAL, the next substance observed during the present study in NRG1-positive enteric cells, shows different effects on the intestinal muscles (similar to SP), depending on the particular fragment of the GI tract and on the precise animal species. For instance, GAL induces muscle contraction in rats and guinea pigs, while in dogs it has relaxatory effects [49,50]. Moreover, GAL (similarly to VIP and NO) inhibits the secretion of some factors in the GI tract [34]. Also noteworthy are the differences in the degree of co-localization of NRG1 with other substances studied during the present investigation, which are best seen by comparing the descending colon with other parts of the large intestine. Such differences strongly suggest that the exact roles of NRG1 in the enteric structures depend on the intestinal fragment, as is the case with other neuronal substances, including, for example, GAL and SP [46,47,48,49,50,51].

Apart from the neurochemical characterization of NRG1-positive enteric neurons under physiological conditions, changes after the administration of BPA were observed. The changes noted in this study, together with those noted in previous publications where the impact of BPA on the total number of NRG1-positive nervous structures was described [25], may suggest the participation of this substance in processes connected with the toxic effects of BPA.

Due to the multidirectional adverse effects of BPA on living organisms, the exact mechanisms of observed changes are difficult to define. They may result from the known direct neurotoxic and neurodegenerative impact of BPA on the nervous system [52], the influence of BPA on the integrity of the intestinal mucosal layer [8], the pro-inflammatory and immunomodulatory activity of BPA [53] or they may be direct effects on the intestinal muscles [54]. Nevertheless, the observed changes are most likely connected with protective and/or adaptive processes which aim to ensure the proper functioning of the ENS and GI tract, as well as the maintenance of homeostasis under conditions changed by the administration of BPA. This is proven by previous studies in which NRG1 was described as an important factor participating in neuroprotective mechanisms and as being necessary for the normal development and functioning of the central and peripheral nervous system [19,20,21,22].

These functions of NRG1 are also confirmed in the current study by co-localization with other substances included in the experiment. It should be emphasized that all neuronal factors whose co-localization with NRG1 were investigated in this study have previously been described as substances involved in the neuroprotective mechanisms. However, the substance that is best known, and which is the strongest in this regard, is VIP, whose neuroprotective activity relies on the influence of the glial cells located around the enteric neurons [55]. Other substances included in the study also show neuroprotective effects. It is known that SP may protect neurons from apoptosis and potassium deprivation-induced cell death [56,57], that GAL enhances the survivability of neurons during brain injury and neurodegenerative diseases [58] and that nNOS influences the synthesis and expression of various factors in the ENS [59]. Given the previous studies describing the involvement of NRG1 in neuroprotective processes and the high degree of co-localization of NRG1 and other neuroprotective substances noted in the present study, it can be said with a high degree of probability that NRG1 in the enteric neurons of the porcine large intestine has neuroprotective functions.

However, it cannot be excluded that the changes observed may be connected with the immunomodulatory and pro-inflammatory activity of BPA, which in the previous studies was described as a factor causing clear changes in the immunological system [53,60,61]. On the other hand, all substances which in the present study were noted in NRG1-positive enteric neurons are involved in the modulation of inflammatory reactions and take part in processes connected with inflammation. Interestingly, it is known that these substances often may act contrary to one another. VIP is an important anti-inflammatory agent and factor maintaining the balance between anti- and pro-inflammatory cytokines [62]. Similarly, anti-inflammatory activity is also shown by GAL [63]. Contrary to VIP and GAL, SP shows pro-inflammatory activity. It influences the receptors located on the surface of immunological cells and causes an increase in the secretion of pro-inflammatory cytokines, including (among others) TNF-α [64,65]. In turn, the functions of NO in the regulation of inflammatory processes is not clear, because this substance may play both anti- and pro-inflammatory factors and the character of the activity depends on the type of inflammation [66,67]. Interestingly, in the present study, the increase in the degree of co-localization of NRG1 with other substances was associated with both anti-inflammatory and pro-inflammatory factors. Such a situation may be connected with the response to the impact of BPA made by the organism to inhibit inflammatory changes (by an increase of the synthesis of anti-inflammatory agents), although it was precisely the impact of BPA that may be the reason for the changes in the level of pro-inflammatory factors.

Of course, it should be pointed out that the present study has some limitations. Firstly, it is not clear what mechanisms are at the root of changes in the neurochemical characterization of NRG1-positive enteric neurons and whether these changes are the result of fluctuations in the expression (protein or mRNA) of the substances studied or of changes in transport within neuronal cells. The effect of the observed changes on the function of the large intestine is also unexplained. Further investigations are indispensable for the clarification of these issues.

In conclusion, the present study, for the first time, has shown that NRG1-positive enteric neurons located in the large intestine are characterized by considerable diversity in terms of neurochemical characterization. Substances which co-localize with NRG1 are involved in various regulatory processes. This fact, together with the presence of NRG1 in all types of enteric plexus, suggests that NRG1 participates in the regulation of diverse intestinal activity, including motility, secretory activity and intestinal and mesenteric blood flow. Moreover, changes in neurochemical characterization of NRG1-positive neurons under the impact of BPA may suggest that these neurons take part in pathological processes connected with the adverse effect of this toxic substance. However, the elucidation of these issues requires further study.

## 4. Materials and Methods

The present study was conducted on 15 immature eight-week-old female Piétrain x Duroc pigs, which were kept under standard laboratory conditions. Twice a day the animals received commercial feed appropriate to their species and age and they were given water ad libitum. All activities carried out during the study received agreement from the Local Ethics Committee in Olsztyn (Poland) (decision numbers 28/2013 of 22 May 2013 and 65/2013/DLZ of 27 November 2013).

After an adaptive period, the animals were divided into three groups of five pigs each: (1) a control (C) group, which received empty capsules; (2) a low dose (LD) group, to which capsules with BPA at a dose of 0.05 mg/kg b.w./day were given; and (3) a high dose (HD) group, in which animals were treated with capsules containing BPA at a dose of 0.5 mg/kg b.w./day. All animals received capsules for 28 days in the same way, i.e., capsules were given orally before the morning feeding. After 28 days, all animals were euthanized. For this purpose, the animals were first pre-medicated with an intramuscular injection of Stresnil (Janssen, Beerse, Belgium, 75 μL/kg of b.w.) and later (after about 30 min) an overdose of sodium thiopental (Thiopental, Sandoz, Kundl, Austria) was given intravenously.

Immediately after euthanasia, the fragments of the selected part of the large intestine were collected. Approximately three-centimetre-long fragments of the caecum (a fragment located 7 cm from the ileocecal valve), the ascending (the apex located between centripetal and centrifugal turns) and descending (fragments, where nerves from the inferior mesenteric ganglia supply the intestine) colon were included in the study. Immediately after collection, the intestinal fragments were put into 4% buffered paraformaldehyde (pH 7.4) for 1 h. The tissues were then rinsed in a phosphate buffer for three days and stored in 18% phosphate-buffered sucrose at 5 °C. After at least three weeks the intestinal fragments were frozen at −22 °C and cut using a cryostat (HM 525, Microm International, Dreieich, Germany). Slices of the intestine were mounted on microscopic slides and stored at −22 °C until further study.

The slices of the intestine underwent the double immunofluorescence technique according to the method described previously [27]. The procedure was as follows: First, after removal from the freezer, the slices were dried at room temperature (rt) for 1 h. Then they were treated with a “blocking” solution consisting of 10% normal goat serum, 0.1% bovine serum albumin, 0.01% NaN_3_, 0.25% Triton x-100 and 0.05% thimerosal in PBS (1 h, rt). The next stage of the labelling was the incubation of intestinal slices with a mixture of two antibodies obtained from various species and directed against neuronal factors studied in a humidity chamber (overnight, rt). One of the antibodies was directed against NRG1, and the second was directed against one of the other substances studied. The list of antibodies is presented in Table 4. On the following day, intestinal fragments were incubated with a mixture of secondary antibodies (humidity chamber, 1 h, rt). Finally, the intestinal slices were treated with buffered glycerol and covered with coverslips. Between each stage of labelling, the intestinal fragments were rinsed in PBS (3 × 10 min.). To verify the specificity of labelling routine tests of antibodies, such as pre-absorption and omission, replacement tests were used.

After labelling, the intestinal fragments were evaluated under an Olympus BX51 microscope using epi-fluorescence and appropriate filter sets. To define the neurochemical characterization of NRG1-positive enteric neurons, at least 400 such neurons located in each type of the enteric plexus in each animal were checked for the presence of each of the other neuronal substances studied (i.e., at least 400 NRG1-positive neurons were tested for the presence of SP, 400 NRG1-positive neurons were tested for the presence GAL, etc.) Only well-stained and visible neurons, whose colour was clearly different from the surrounding area, were included in the study. The neurochemical characterization of NRG1-positive neurons was presented as a percentage of neurons containing SP, GAL, VIP or nNOS in relation to the total number of NRG1-like immunoreactive (LI) cells, which was considered as 100%. For example, when, during the co-localization study of NRG1 and SP, 400 cells immunopositive to NRG1were evaluated and the presence of SP was noted in 100 cells, the value was presented as 25%. In order to exclude the possibility of the same cells being double-counted, the intestinal slices included in the study were located at least 200 µm apart from each other. The data were pooled and presented as the mean ± SEM. A statistical analysis was performed with a one-way analysis of variance (ANOVA), with Bonferroni’s Multiple Comparison post hoc test using Statistica 12 software (StatSoft Inc., Tulsa, OK, USA). Differences were considered significant at *p* < 0.05 (*) and as highly significant at *p* ≤ 0.01 (**) and *p* ≤ 0.001 (***).

## Figures and Tables

**Figure 1 ijms-22-10308-f001:**
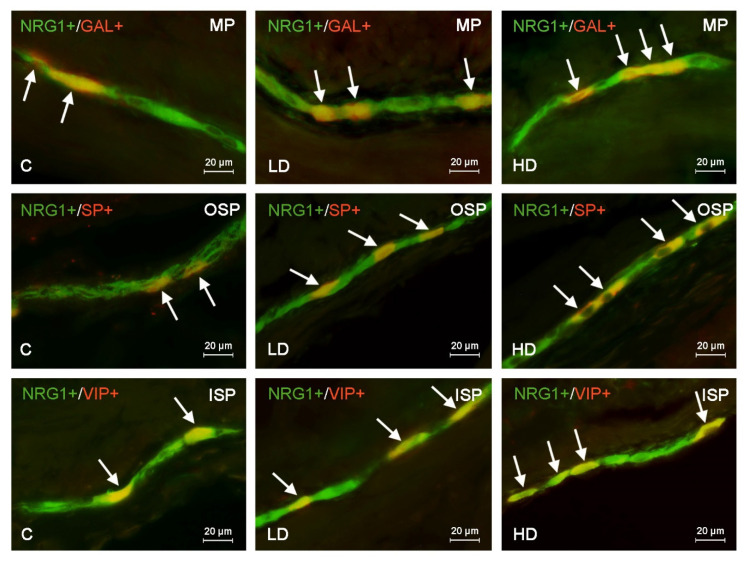
Distribution pattern of nerve cells immunoreactive to Neuregulin 1 (NRG1) and other neuronal substances studied—galanin (GAL), substance P (SP) or vasoactive intestinal polypeptide (VIP)—in the muscular plexus (MP), outer submucous plexus (OSP) or inner submucous plexus (ISP) in the porcine caecum under physiological conditions (C) and under the impact of a low dose (LD) and a high dose (HD) of BPA. The pictures are the result of the overlap of both stainings. Neurons immunoreactive for both NRG1 and other of the neuronal substances studied are indicated with arrows.

**Figure 2 ijms-22-10308-f002:**
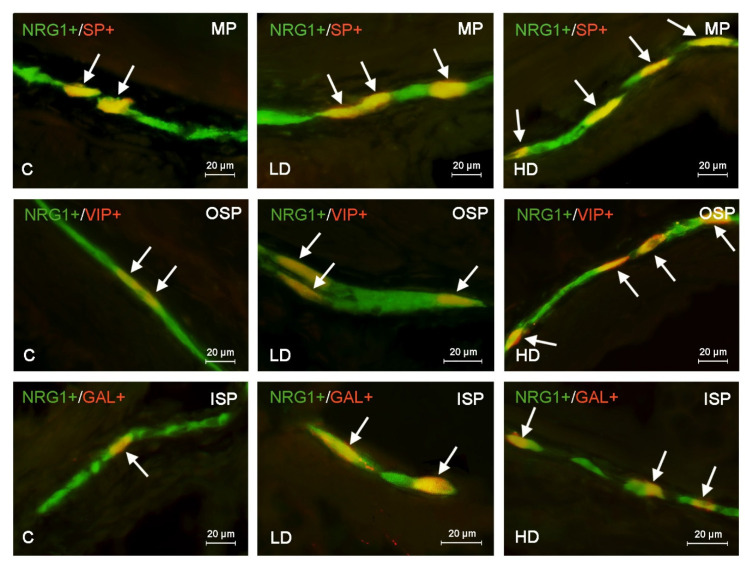
Distribution pattern of nerve cells immunoreactive to Neuregulin 1 (NRG1) and other neuronal substance studied—galanin (GAL, substance P (SP) or vasoactive intestinal polypeptide (VIP)—in the muscular plexus (MP), outer submucous plexus (OSP) or inner submucous plexus (ISP) in the porcine ascending colon under physiological conditions (C) and in the low dose (LD) group and the high dose (HD) group. The pictures are the result of the overlap of both stainings. Neurons immunoreactive for both NRG1 and other of the neuronal substances studied are indicated with arrows.

**Figure 3 ijms-22-10308-f003:**
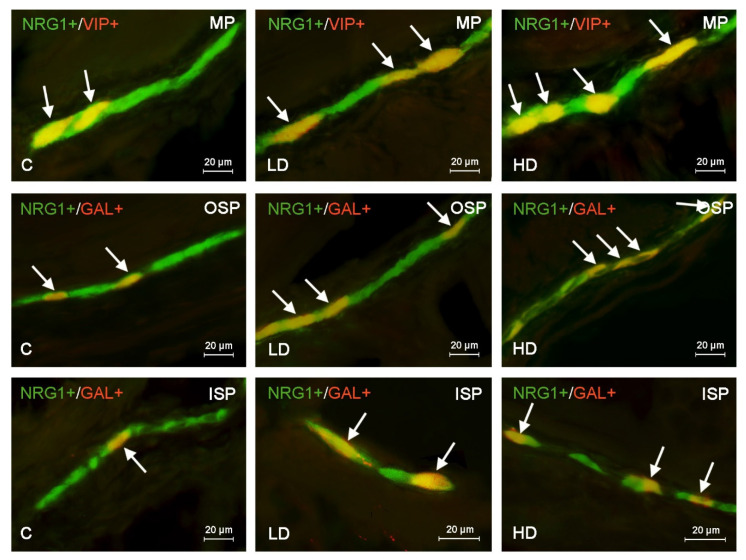
Distribution pattern of nerve cells immunoreactive to Neuregulin 1 (NRG1) and other neuronal substances studied—galanin (GAL), substance P (SP) or vasoactive intestinal polypeptide (VIP)—in the muscular plexus (MP), outer submucous plexus (OSP) or inner submucous plexus (ISP) in the porcine descending colon under physiological conditions (C) and in the low dose (LD) group and the high dose (HD) group. The pictures are the result of the overlap of both stainings. Neurons immunoreactive for both NRG1 and other of the neuronal substances studied are indicated with arrows.

**Table 1 ijms-22-10308-t001:** Co-localization of NRG1 with other neuronal active substances under physiological conditions (C group) and after BPA administration in the low dose (LD) group and high dose (HD) group in caecum.

Part of ENS	Experimental Group	Active Neuronal Substances Studied during the Experiment			
		SP	GAL	VIP	NOS
Myenteric plexus	C group	21.05 ± 0.38%	19.94 ± 0.20%	26.20 ± 0.33%	31.67 ± 0.73%
	LD group	28.77 ± 0.28% ***	28.06 ± 0.25% ***	32.23 ± 0.16% ***	35.96 ± 0.33% ***
	HD group	36.59 ± 0.34% ***	35.44 ± 0.18% ***	36.56 ± 0.22% ***	40.67 ± 0.50% ***
Outer submucous plexus	C group	23.47 ± 0.37%	21.43 ± 0.27%	26.39 ± 0.29%	32.06 ± 0.58%
	LD group	26.75 ± 0.19% ***	26.99 ± 0.24% ***	32.62 ± 0.19% ***	34.30 ± 0.25% **
	HD group	33.79 ± 0.26% ***	30.22 ± 0.31% ***	35.89 ± 0.29% ***	39.61 ± 0.24% ***
Inner submucous plexus	C group	22.14 ± 0.42%	22.00 ± 0.31%	24.93 ± 0.30%	30.46 ± 0.21%
	LD group	33.07 ± 0.41% ***	28.27 ± 0.19% ***	32.45 ± 0.29% ***	32.67 ± 0.35% ***
	HD group	38.05 ± 0.23% ***	30.84 ± 0.27% ***	39.09 ± 0.12% ***	37.97 ± 0.25% ***

NRG1-positive neurons were considered as representing 100% for all combinations with selected neuronal factors, and all the values presented are percentages (means ± SEM) of NRG1-positive enteric neurons. Highly statistically significant (*p* ≤ 0.01 and *p* ≤ 0.001) differences between the C group and LD group, as well as the C group and HD group, are marked with ** and ***, respectively.

**Table 2 ijms-22-10308-t002:** Co-localization of NRG1 with other neuronal active substances under physiological conditions (C group) and after BPA administration in the low dose (LD) group and high dose (HD) group in ascending colon.

Part of ENS	Experimental Group	Active Neuronal Substances Studied during the Experiment			
		SP	GAL	VIP	NOS
Myenteric plexus	C group	22.84 ± 0.28%	17.83 ± 0.38%	24.00 ± 0.33%	32.80 ± 0.84%
	LD group	30.83 ± 0.29% ***	22.21 ± 0.29% ***	30.07 ± 0.34% ***	37.88 ± 0.87% **
	HD group	36.30 ± 0.25% ***	32.58 ± 0.21% ***	35.34 ± 0.28% ***	43.09 ± 0.93% ***
Outer submucous plexus	C group	22.12 ± 0.33%	17.80 ± 0.26%	35.76 ± 0.36%	33.58 ± 0.76%
	LD group	30.27 ± 0.39% ***	25.35 ± 0.41% ***	43.02 ± 0.25% ***	35.25 ± 0.48%
	HD group	39.64 ± 0.29% ***	32.87 ± 0.26% ***	47.65 ± 0.30% ***	40.26 ± 0.30% ***
Inner submucous plexus	C group	21.82 ± 0.43%	18.53 ± 0.25%	21.39 ± 0.23%	31.52 ± 0.49%
	LD group	29.78 ± 0.83% ***	24.82 ± 0.18% ***	36.09 ± 0.40% ***	34.61 ± 0.62% **
	HD group	36.03 ± 0.13% ***	30.67 ± 0.29% ***	45.74 ± 0.24% ***	39.23 ± 0.27% ***

NRG1-positive neurons were considered as representing 100% for all combinations with selected neuronal factors, and all the values presented are percentages (means ± SEM) of NRG1-positive enteric neurons. Highly statistically significant (*p* ≤ 0.01 and *p* ≤ 0.001) differences between the C group and LD group, as well as the C group and HD group, are marked with ** and ***, respectively.

**Table 3 ijms-22-10308-t003:** Co-localization of NRG1 with other neuronal active substances under physiological conditions (C group) and after BPA administration in the low dose (LD) group and high dose (HD) group in descending colon.

Part of ENS	Experimental Group	Active Neuronal Substances Studied during the Experiment			
		SP	GAL	VIP	NOS
Myenteric plexus	C group	25.54 ± 0.45%	30.14 ± 0.69%	27.94 ± 1.82%	34.17 ± 1.66%
	LD group	25.76 ± 0.38%	37.71 ± 0.34% ***	34.90 ± 0.88% *	39.41 ± 1.43%
	HD group	31.23 ± 0.66% ***	47.20 ± 0.91% ***	42.92 ± 1.90% ***	43.25 ± 1.62% **
Outer submucous plexus	C group	29.11 ± 2.28%	38.33 ± 1.06%	31.74 ± 0.73%	35.60 ± 1.11%
	LD group	30.49 ± 0.92%	44.26 ± 1.14% **	35.38 ± 0.78% *	36.58 ± 0.30%
	HD group	35.29 ± 0.61% *	48.67 ± 0.49% ***	40.15 ± 1.09% ***	40.80 ± 0.55% ***
Inner submucous plexus	C group	30.71 ± 0.90%	38.70 ± 1.31%	32.39 ± 1.95%	32.15 ± 1.07%
	LD group	32.36 ± 1.29%	44.94 ± 0.78% *	37.67 ± 1.02%	35.53 ± 0.78% *
	HD group	37.60 ± 1.54% **	49.09 ± 1.83% ***	40.02 ± 1.08% **	39.44 ± 0.54% ***

NRG1-positive neurons were considered as representing 100% for all combinations with selected neuronal factors, and all the values presented are percentages (means ± SEM) of NRG1-positive enteric neurons. Statistically significant (*p* ≤ 0.05) and highly statistically significant (*p* ≤ 0.01 and *p* ≤ 0.001) differences between the C group and LD group, as well as the C group and HD group, are marked with *, ** and ***, respectively.

**Table 4 ijms-22-10308-t004:** The list of antisera and reagents used in immunohistochemical investigations.

Primary Antibodies				
Antigen	Code	Species	Working Dilution	Supplier
NRG 1	AA 198–229	Rabbit	1:1000	Antibodies-online, Aachen, Germany
VIP	9535-0504	Mouse	1:1000	Biogene, Kombolton UK
SP	8450-0505	Rat	1:1000	BioRad, Hercules, CA, USA
nNOS	N218	Mouse	1:1000	Sigma-Aldrich, Saint Louis, MO, USA
GAL	T-5036	Guinea Pig	1:2000	Peninsula Labs, San Carlos, CA, USA
**Secondary Antibodies**				
**Reagents**			**Working Dilution**	**Supplier**
Alexa Fluor 488 donkey anti-rabbit IgG			1:1000	ThermoFisher Scientific Waltham, MA, USA
Alexa Fluor 546 donkey anti-mouse IgG			1:1000	ThermoFisher Scientific
Alexa Fluor 546 donkey anti-rat IgG			1:1000	ThermoFisher Scientific
Alexa Fluor 546 donkey anti-guinea pig IgG			1:1000	ThermoFisher Scientific

## Data Availability

Data is contained within the present article.

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
