# Peer review of "The Influence of Bisphenol A (BPA) on the Occurrence of Selected Active Substances in Neuregulin 1 (NRG1)-Positive Enteric Neurons in the Porcine Large Intestine"

_ijms, 2021, doi:10.3390/ijms221910308_

Round 1
Reviewer 1 Report
This paper addresses the impact of BPA for neuregulin1 (NRG1), which is expressed in the enteric nervous system (ENS). The authors indicated that NRG1-positive neurons may contain substance P, vasoactive intestinal polypeptide, neuronal isoform of nitric oxide synthase and galanin, and exposure of BPA accelerated the co-localization of NGR1 and these biomolecules. The paper mentioned a new effect of BPA on intestine; however, their experiments are insufficient and somehow unclear to support their results and discussion. As for co-localization analysis, the threshold level applied were unclear. It is strongly recommended to analyze the expression levels of mRNAs of NRG1 and all other target molecules to analyze the impact of exposure of BPA on intestine.
Author Response
The authors thank the Reviewer for evaluation of the manuscript.
It should be underlined that the subject of the experiment was the co-localization of NRG1 with other neuronal active substances in the enteric neurons and the influence of BPA on this co-localization. It is the first characterization of chemical profile of NRG1-positive enteric neurons in physiological conditions and after BPA administration. The study was not intended to determine the effect of BPA on synthesis of neuronal substances in the whole intestinal wall. It should be pointed out that method proposed by the Reviewer is of course novelty and precise and very useful in many types of studies, but it is not suitable to describe changes in co-localization of substances in the same nervous structures. This method is even not suitable to describe changes in the enteric nervous system, because a lot of substances are synthetized not only in the neurons, but also in other types of cells located in the intestinal wall (for example in enteroendocrine cells, enterocytes, glial cells). Therefore evaluation of mRNA levels can show changes in the whole intestine wall, not changes in the enteric neurons, and certainly not the neurochemical characterization of enteric neurons and degree of colocalization of substances in the same nervous structures (what was the aim of study). Moreover, the experiment has been ended and additional experimental determinations are not possible without new experimental animals and new agreement of ethical approval. The authors have hope that mentioned above explanation that the subject of the study did not require suggested experimental methods, will convince the Reviewer.
Moreover materials and methods has been reedited to more precise description of methods used in the study. The authors are also in agreement that some conclusions are not supported by the results. Therefore discussion has also been reedited.
Reviewer 2 Report
In the current manuscript, the authors present data about the co-localization of neuregulin 1 with several other proteins in the enteric nervous system of the porcine large intestine under physiological conditions and after administration of bisphenol A. They found a general increase in the degree of this co-localization after bisphenol A.
Obviously, this study is a continuation of the work published in the same journal 2018. At that time, the authors studied the influence of bisphenol A on some other proteins in the ileum and now, they had a look on the colon. Moreover, I would assume that the experimental animals (15 female pigs) were the very same despite the provided ethical decision numbers are different, but from the same year.
With regard to the methods and to the previous work, this study appears as a consistent approach. Moreover, the influence of bisphenol A on the human health is a highly relevant topic. However, in this context, the manuscript is somewhat disappointing as it is lacking any functional or mechanistical data about the effects of bisphenol A. In the study in 2018, in my view, the authors promised to investigate also some mechanistical aspects of bisphenol A in their future work. This was clearly not the case. The current work is purely descriptive and the conclusions are, in my view, of no significance: 1) It is completely unclear how bisphenol A contributes to the observed results. 2) It is completely unclear if the expression (protein or mRNA) of the studied proteins is changed or if it was a redistribution process in the neurons. 3) It is completely unclear if the function of the enteric nervous system or of the colon is affected. 4) It is completely unclear if the observed bisphenol A-dependent higher degree of co-localization of several neuronal proteins is protective, adaptive, worse, or without significance.
In summary, the manuscript is of low priority in the current form. I would recommend to add some mechanistical data. If this is not possible, a detailed limitation of the study should be provided and the last sentence of the abstract (lines 24-26) should be omitted because the results do not support this conclusion.
Author Response
The authors thank the reviewer very much for review. Of course the authors are in agreement with the Reviewer that a lot of questions connected with the influence of BPA on NRG1-positive enteric neurons are still not clear. Moreover the authors are aware of the limitations of the manuscript mentioned by the Reviewer. According to suggestions of the Reviewer these limitations have been added to discussion. Moreover the Reviewer is right that some conclusions are not supported by the results, and so the abstract and discussion have been reedited according to suggestions of the Reviewer.
On the other hand it should be underlined that the study was aimed at the description of colocalization of NRG1 with other substances in the same enteric neurons under physiological conditions and after BPA administration. Method suggested by the Reviewer (evaluation of mRNA levels) is of course novelty and useful method, but it is not right to evaluation of neurochemical characterization of enteric neurons and colocalization of substances in the same structures. This method allows to observed changes in the synthesis of substances in the whole intestinal wall, not changes in the enteric nervous system, because the same substances are synthetized not only by neurons, but also by other cells located in the intestinal wall (enterocytes or endocrine cells). The best method to show colocalization of neuronal substances in the same nervous structures is method used in the manuscript. Moreover, unfortunately the experiment has been ended and new experimental evaluations are not possible without new experimental activities (new animals, new ethical approvement). Therefore, the authors have hope that mentioned above explanation will convince the Reviewer that evaluation of mRNA levels is not necessary for the purpose of the research (namely the description of the co-location of NRG1 with other substances in the intestinal neurons).
Round 2
Reviewer 1 Report
The authors did not provide additional experimental data, but they added the description as for the limitation of their research results.
Reviewer 2 Report
The discussion of the manuscript has been reedited to my satisfaction.